# A Possibility of Vasospastic Angina after mRNA COVID-19 Vaccination

**DOI:** 10.3390/vaccines10121998

**Published:** 2022-11-24

**Authors:** Toru Awaya, Masao Moroi, Fuminori Nakamura, Satoru Toi, Momoko Wakiya, Yoshinari Enomoto, Taeko Kunimasa, Masato Nakamura

**Affiliations:** Department of Cardiovascular Medicine, Toho University Ohashi Medical Center, 2-22-36 Ohashi Meguro-ku, Tokyo 153-8515, Japan

**Keywords:** COVID-19 vaccination, cardiovascular diseases, vasospastic angina, adverse reaction, inflammatory cytokine, lipid nanoparticles, angiotensin-converting enzyme 2, alcohol

## Abstract

We report a case of vasospastic angina (VSA) following COVID-19 mRNA vaccination. Despite the widespread occurrence of myocarditis, there have been few reports of post-vaccinal VSA. A 41-year-old male patient was referred for chest pain at rest following mRNA vaccination; he had never experienced chest pain prior to vaccination. He was diagnosed by an acetylcholine (Ach) provocation test that showed multivessel vasospasm. After the initiation of treatment with a calcium channel blocker and nitrate, no further exacerbation of chest pain was observed. To our knowledge, this constitutes the first reported case of VSA proven by Ach provocation test after COVID-19 vaccination. The vaccination may increase coronary artery spasticity. VSA should be ruled out in post-vaccine new onset resting chest pain.

## 1. Introduction

Messenger RNA (mRNA)-based vaccination against COVID-19 is used to combat the spread of the SARS-CoV-2 virus. Cardiovascular diseases, including acute coronary syndrome (ACS) and myocarditis, are the most serious side effects of COVID-19 mRNA vaccination [1,2]. Plausible causes of ACS following vaccination are assumed to be angiotensin-converting enzyme 2 (ACE2) downregulation [3,4,5], inflammatory cytokine [6,7,8,9,10,11], and allergic response [12,13]. Although ACS has been reported many times, vasospastic angina (VSA) is rarely reported after the COVID-19 mRNA vaccine.

## 2. Case Presentation

A 41-year-old Japanese male smoker with no medical history received the first dose of the mRNA-BNT162b2 COVID-19 vaccine (Pfizer-BioNTech, Cambridge, MA, USA). One week after vaccination, chest pain and palpitations appeared after drinking alcohol; he had no previous documented chest pain prior to his COVID-19 vaccination. His chest pain repeated exacerbations and remissions at rest and he was admitted to our hospital on the 11th day after vaccination.

On admission, his body temperature was 37.2 °C, blood pressure was 126/80 mmHg, heart rate was 98/min, and oxygen saturation was 98% on room air. When he arrived at our hospital, his chest pain was resolved, and his electrocardiogram (ECG) and transthoracic echocardiogram were normal. His blood test results were within normal limits, including creatine kinase (124 IU/L), high-sensitivity cardiac troponin T (0.003 ng/mL), potassium (3.7 mEq/L), creatinine (0.83 mg/dL), and C-reactive protein (0.04 mg/dL) on admission. After he developed chest pain at rest and after alcohol consumption, we suspected VSA. Emergency coronary angiography revealed no coronary artery stenosis. However, the acetylcholine (Ach) provocation test showed multivessel vasospasm with chest pain (left coronary artery (LCA), segments #7 99% stenosis and #12 total occlusion at 100 µg dose into LCA) with ST elevation in ECG leads I, aVL and large T wave in leads V1–V4 (Figure 1A–E). He was diagnosed with VSA because his findings fulfilled all criteria (i) reproduction of the usual chest pain, (ii) ischemic ECG changes, and (iii) 90% vasoconstriction on angiography.

Therefore, we initiated a calcium channel blocker (CCB) with nitrate and he was discharged the next day. Antibody titers of various viruses, including SARS-CoV-2, were negative. After the initiation of treatment with a CCB and nitrate, no further exacerbation of chest pain was observed.

## 3. Discussion

To our knowledge, this constitutes the first reported case of VSA proven by Ach provocation test after COVID-19 vaccination. He was diagnosed with VSA because his Ach provocation test findings fulfilled all criteria: (i) reproduction of the usual chest pain, (ii) ischemic ECG changes, and (iii) 90% vasoconstriction on angiography [14]. Additionally, after the initiation of treatment with a CCB and nitrate, no further exacerbation of chest pain was observed.

Chest discomfort is present in 3% of cases after receiving the mRNA vaccine [15]. Myocarditis, acute coronary syndrome, and VSA have been reported as potential causes of chest pain associated with mRNA vaccination against COVID-19 [1,2]. However, reported cases of VSA following the COVID-19 mRNA vaccine are scarce and the true incidence remains unknown. Acute coronary syndromes (ACS) and angina pectoris in the elderly are reported to often present as atypical chest pain and more cardiac failure [16,17]. Therefore, it is important not to rule out ACS or angina even in the presence of atypical chest pain after vaccination in the elderly. Notably, a flowchart of the differential diagnoses of chest discomfort and palpitation after COVID-19 vaccination has been reported [18].

Regarding the relationship between vaccination and VSA, several mechanisms may induce VSA. The spike protein of the SARS-CoV-2 virus or the mRNA component of the vaccine binds to the angiotensin-converting enzyme 2 (ACE2) receptor, which leads to the downregulation of ACE2 (Figure 2) [3,4]. ACE2 converts the potent vasoconstrictor angiotensin II (AngII) to the vasodilator angiotensin 1–7. Hence, the downregulation of ACE2 activates AngII, which leads to vasoconstriction and also produces interleukin (IL)-6 [3,5]. Various inflammatory stimuli, including AngII and IL-6, upregulate the activity of Rho-kinase. The activation of Rho-kinase plays a central role in coronary artery spasms caused by the hypercontraction of vascular smooth muscle cells (Figure 2) [19].

Other plausible mechanisms are the involvement of inflammatory cytokines and allergic reactions. While lipid nanoparticles (LNPs) play an important role as a delivery system for mRNA vaccines, cationic lipids in LNPs themselves reportedly cause inflammatory cytokines such as interleukin (IL)-6 and IL-1β (Figure 2) [7,8,9]. The IL6 genotype has been reported to affect the prevalence of VSA [10]. Four post-mRNA vaccine autopsy cases have recently been reported to express genes that upregulate inflammatory cytokine signaling compared with controls [6]. Inflammatory cytokines inhibit endothelial nitric oxide and increase Rho-kinase activity, causing coronary artery spasms (Figure 2) [11,19].

East Asians are genetically predisposed to VSA since 40% are aldehyde dehydrogenase 2 (ALDH2)*2 carriers with deficient ALDH2 activity [20]. In ALDH2*2 carriers, alcohol intake and tobacco smoking increase toxic aldehydes, which cause endothelial dysfunction (Figure 2). Although we did not examine the ALDH2 genotype of the patient, we speculated that he was an ALDH2*2 carrier since he had alcohol flushing syndrome (AFS). AFS is associated with ALDH2*2 (odds ratio 51.6) [20]. He had never experienced chest pain prior to receiving the COVID-19 mRNA vaccine; therefore, post-vaccinal alcohol consumption may be a factor that triggered VSA.

COVID-19 mRNA vaccine components, including polyethylene glycol, LNPs, etc., may cause an allergic reaction, resulting in a coronary vasospasm called Kounis syndrome [2,12]. Mast cells activated by the vaccine antigen induce histamine with or without immunoglobulin E [12], and histamine reportedly induces coronary vasospasm [13]. β1-adrenoceptors (ARs) are predominantly found in the coronary epithelium, while β2-ARs are found in the microvascular [21]. It has recently been reported that Kounis syndrome is more likely to occur when β2-ARs, but not β1, are blocked [22]. Therefore, patients taking nonselective β-ARs antagonists such as propranolol should be aware of Kounis syndrome following vaccination.

Although this case highly suggested the relationship between COVID-19 mRNA vaccine and VSA, VSA and microvascular dysfunction, not epicardial coronary artery stenosis has also been reported in COVID-19 patients [23,24]. Therefore, the benefit–risk balance should always be considered when deciding whether or not to vaccinate.

## 4. Conclusions

This case demonstrates a possibility of VSA after COVID-19 mRNA vaccination. VSA should be ruled out in post-vaccine new onset resting chest discomfort and palpitation, especially in cases with repeated exacerbations and remissions.

## Figures and Tables

**Figure 1 vaccines-10-01998-f001:**
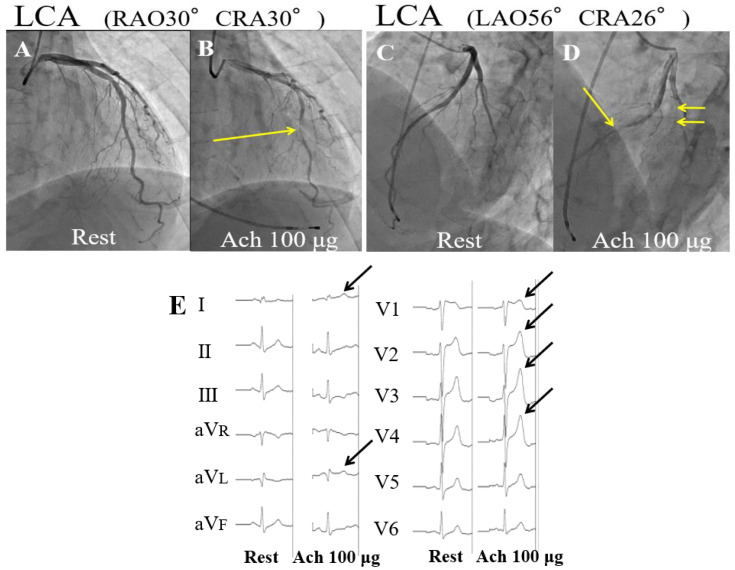
(**A**,**C**): Coronary angiography (CAG) revealed a nonobstructive coronary artery on admission. (**B**,**D**): An acetylcholine (Ach) provocation test revealed multivessel vasospasm (left coronary artery (LCA) segments, #7 99% stenosis and #12 total occlusion (yellow arrows) at 100 µg dose into LCA). (**E**): An electrocardiogram showed ST elevation in leads I, aVL, and large T wave in leads V1–V4 (black arrows) during the Ach provocation test.

**Figure 2 vaccines-10-01998-f002:**
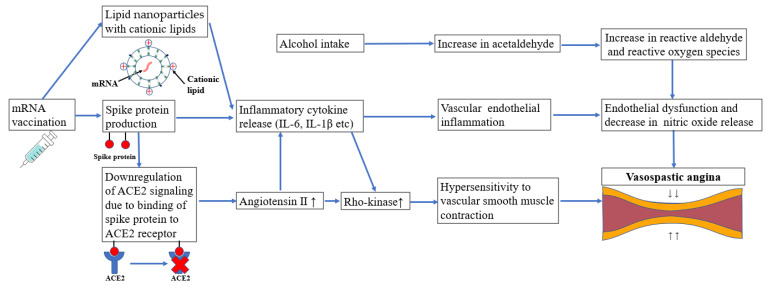
The potential mechanisms of vasospastic angina following mRNA vaccination in this case. mRNA, messenger RNA; IL, interleukin; ACE2, angiotensin-converting enzyme 2.

## Data Availability

Data sharing is not applicable to this article as no datasets were generated or analyzed during the current study.

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
