# Peer review of "A Possibility of Vasospastic Angina after mRNA COVID-19 Vaccination"

_vaccines, 2022, doi:10.3390/vaccines10121998_

Round 1

Reviewer 1 Report

I would congratulate for this very interesting case demonstrating  that VSA should be ruled out in post-vaccine new-onset resting chest discomfort , especially in cases with repeated exacerbations and remissions. Myocarditis, acute coronary syndrome, have been reported as potential causes of chest pain associated with mRNA vaccination against COVID-19  However, reported cases of VSA following the COVID-19 mRNA vaccine are scarce and this case report is something important. Here you find 3 points in order to improve the case

1.In discussion authors should more discuss the impact of the age in this case report, since elderly patients may be potentially show atypical presentation with less chest pain and more cardiac failure. Please amplify discussion cite 2 important references: DOI: 10.4081/monaldi.2006.537 ; and doi: 10.2459/JCM.0000000000000603

2.In discussion authors should discuss about a risk-benefit balance where a benefit-risk evaluation based on the potential benefits should outweigh the potential risks: in particular, also cases of angina post COVID infection have been described (DOI: 10.1016/j.rec.2021.10.010; and DOI: 10.2169/internalmedicine.0137-22). Please amplify this fundamental discussion and cite 2 suggested reference

3. Beta-adrenergic vasodilation plays a key role in the regulation of coronary blood flow, in particular during sympathetic nervous system activation. Beta1-adrenergic receptors are mostly represented in epicardial coronary vessel, while beta2 receptors are located in microvasculature (Recenti Prog Med. 2005 Sep;96(9):411-5). Please discuss this point focusing on potential relation with vaccine, and cite the suggested reference

4. A nice figure summarizing all potential mechanism involved in the angina post-vaccination is absolutely welcome for readers

5.Please update reference list with 5 suggested papers

Reviewer 2 Report

Authors have correctly demonstrated the presence of VSA after mRNA COVID-19 vaccination. The assumption that VSA is caused by a Vaccine is only at the level of possibility, which the authors acknowledge in the discussion and conclusion. However, the title of the paper may give the impression that the VSA is actually caused by mRNA COVID-19 vaccination. It would be more correct to mark vaccination only as a possibility of VSA in the title.

Author Response

Thank you for your positive feedback our manuscript. We reflected your suggestion into our title.

Title

A case of vasospastic angina after mRNA COVID-19 vaccination→

A possibility of vasospastic angina after mRNA COVID-19 vaccination

Reviewer 3 Report

This is interesting case report of vasospastic angina following COVID-19 mRNA vaccination.

I have no major comments. I suggest adding more baseline lab results incl K, Mg, renal function, CRP if available.

This is case report presentation. This is not original article.
Topic is actual and interesting
Case report is well prepared
Text is easy to read
That is why in my opinion is worth to be published after just minor review.

Author Response

We thank the reviewer very much for your positive feedback on our manuscript. We reflected your suggestion into our text. Unfortunately, Mg was not measured on admission; K, Cr, and CRP are additionally listed.

His blood test results were within normal limits, including creatine kinase (124 IU/l), high-sensitivity cardiac troponin T (0.003 ng/mL), potassium (3.7 mEq/L), creatinine (0.83 mg/dl), and C-reactive protein (0.04 mg/dL) on admission.

Round 2

Reviewer 1 Report

I would congratulate with authors. Manuscript definitely improved. No more comments from this reviewer